# USP25 Regulates EGFR Fate by Modulating EGF-Induced Ubiquitylation Dynamics

**DOI:** 10.3390/biom10111548

**Published:** 2020-11-13

**Authors:** Carlos A. Niño, Nadine Wollscheid, Giovanni Giangreco, Elena Maspero, Simona Polo

**Affiliations:** 1IFOM, Fondazione Istituto FIRC di Oncologia Molecolare, 20139 Milan, Italy; Nadine.Wollscheid@gmx.de (N.W.); elena.maspero@ifom.eu (E.M.); 2Tumour Cell Biology Laboratory, The Francis Crick Institute, London NW1 1AT, UK; giovanni.giangreco@crick.ac.uk; 3DIPO, Dipartimento di Oncologia ed Emato-Oncologia, Università degli Studi di Milano, 20122 Milan, Italy

**Keywords:** EGFR, ubiquitin, ubiquitylation, deubiquitylating enzymes, cancer

## Abstract

Deregulated epidermal growth factor receptor (EGFR) signaling is a key feature in different stages of oncogenesis. One important mechanism whereby cancer cells achieve increased and uncontrolled EGFR signaling is escaping down-modulation of the receptor. Ubiquitylation of the EGFR plays a decisive role in this process, as it regulates receptor internalization, trafficking and degradation. Deubiquitinating enzymes (DUBs) may oppose the ubiquitylation process, antagonizing or even promoting receptor degradation. Here, we use qualitative and quantitative assays to measure EGFR internalization and degradation after Ubiquitin Specific Peptidase 25 (USP25) depletion. We show that, by acting at the early steps of EGFR internalization, USP25 restrains the degradation of the EGFR by assisting in the association of the E3 ubiquitin ligase c-Cbl with EGFR, thereby modulating the amplitude of ubiquitylation on the receptor. This study establishes USP25 as a negative regulator of the EGFR down-modulation process and suggests that it is a promising target for pharmacological intervention to hamper oncogenic growth signals in tumors that depend on the EGFR.

## 1. Introduction

The epidermal growth factor receptor (EGFR) belongs to the ErbB family of Receptor Tyrosine Kinases (RTKs) and exerts a fundamental role in development, tissue wound healing and organ physiology [1,2]. Activation of EGFR signaling occurs when a cognate ligand(s) (i.e., EGF) binds to the extracellular EGFR domain, inducing receptor dimerization and activation of its intracellular tyrosine kinase domain which, in turn, triggers a downstream signaling cascade [1]. Uncontrolled EGFR signaling promotes tumorigenesis and is a hallmark of different types of cancers, including glioblastoma, lung, colorectal, pancreatic, and head and neck carcinomas [3]. The aberrant signaling can be a consequence of EGFR and/or ligand(s) overexpression, EGFR mutations that cause spontaneous dimerization or activation, alterations in the EGFR signaling cascade, or sustained signaling caused by inefficient EGFR degradation [4,5,6].

Endocytosis has a pivotal role in fine-tuning EGFR signaling. EGF stimulation induces EGFR internalization through two major pathways, namely clathrin-mediated endocytosis (CME) and non-clathrin endocytic pathway (NCE), resulting in different EGFR fates and signal extension [6]. The balance between recycling and degradation depends on the EGFR ubiquitylation status, which invariably routes the activated receptor from the endosome to the lysosome, where it is degraded [6,7]. Ubiquitylation of the EGFR by the E3 ligase c-Cbl is dispensable for CME [8,9], while it is required for the activation of the NCE pathway [10,11,12]. As a consequence, NCE represents the main mechanism for signal attenuation to protect cells from overstimulation.

At the molecular level, EGFR ubiquitylation is threshold-controlled and depends on the concomitant phosphorylation of a direct c-Cbl binding site (Y1045, [13]) and two indirect sites (Y1068 and Y1086, responsible for GRB2 binding, [14,15]) in the EGFR tail [11,12]. c-Cbl-mediated EGFR ubiquitylation is finely tuned along the endocytic pathway by the counteracting activities of deubiquitinating enzymes (DUBs) [16,17]. Few DUBs, namely AMSH, USP2 and USP22, have been reported to affect EGFR fate at the level of the sorting endosomes by acting directly on the receptor and protecting it from degradation [18,19,20,21]. On the other hand, OTUD7/Cezanne-1 opposes directly c-Cbl-mediated EGFR ubiquitylation at the plasma membrane and enhances EGFR signaling [22]. Other DUBs, such as USP9X and UBPY/USP8, indirectly affect EGFR trafficking and turnover by acting on the endocytic machinery. USP9X modulates Eps15 monoubiquitylation status, which is critical for EGFR internalization [23], whereas UBPY/USP8 stabilizes Hrs and STAM, two components of the endosomal sorting complexes required for transport-0 (ESCRT-0) complex [24,25]. In both cases, depletion of USP9X or UBPY/USP8 results in the accumulation of the EGFR in intracellular endosomal compartments, eluding its degradation.

We previously undertook a systematic RNAi-based approach to dissect the involvement of DUBs in the EGFR pathway and identified 13 DUBs with an unprecedented impact on EGFR degradation kinetics [23]. Here, we report the characterization of USP25, a member of the ubiquitin-specific protease (USP) family that works as a negative regulator of EGFR degradation. By modulating EGFR ubiquitylation, USP25 impacts receptor internalization and degradation, representing a suitable “druggable” target to be further explored in the development of targeted therapies for the treatment of EGFR-dependent tumors.

## 2. Materials and Methods

### 2.1. Reagents and Constructs

EGF was from PeproTech (Cranbury, NJ, USA); Alexa555-EGF (E-35350) was from Molecular Probes (Eugene, OR, USA); 125I-EGF was from Perkin Elmer (Waltham, MA, USA). MG132 (474790) and Chloroquine (C6628) were from SIGMA (MERCK, Darmstadt, Germany). Antibodies (biochemistry) were: rabbit polyclonal anti-EGFR (806, epitope: aa 1172-1186, homo sapiens generated in-house [26]); rabbit polyclonal anti-USP25 (home-made, directed against human full-length GST-USP25); mouse anti-Ubiquitin (ZTA10, [27]). Mouse anti-c-Cbl (610442) was from BD Bioscience (BD Pharmingen, CA, USA). Rabbit anti-GRB2 (sc-255); mouse anti-pY (sc-7020) and mouse anti-GAPDH (sc-32233) were from Santa Cruz (Dallas, TX, USA). Mouse anti-tubulin (T5168) and rabbit anti-GFP (G1544) were from SIGMA (MERCK, Darmstadt, Germany). Mouse anti-pY(1068)EGFR (2236) and rabbit anti-pY(1045)EGFR (2237) were from Cell Signaling (Danvers, MA, USA). Antibodies used in immunofluorescence were: mouse anti-EGFR (13A9, kindly provided by GENENTECH, San Francisco, CA, USA), and Alexa488-conjugated donkey anti-mouse (A21202) Thermo Fisher (Thermo Fisher Scientific, Waltham, MA, USA). Antibodies used in immunoprecipitation studies were: rabbit polyclonal anti-EGFR (#806, epitope: aa 1172-1186, homo sapiens generated in-house [26]), rabbit polyclonal anti-Eps15 (#861 generated in house) and rabbit anti-IgG antibody (I5006), SIGMA (MERCK Darmstadt, Germany).

GFP-tagged human USP25 WT was kindly provided by Sylvie Urbé (University of Liverpool, UK). GFP-tagged USP25C178A was generated by site-directed mutagenesis starting from the wild-type construct with the following oligonucleotides: USP25C178Afwd: CTAAAGAATGTTGGCAATACTGCTTGGTTTAGTGCTGTTATTC and USP25C178Arev: GAATAACAGCACTAAACCAAGCAGTATTGCCAACATTCTTTAG.

Doxycycline-inducible stable cell lines to knock-down USP25 were generated using the pSLIK system. The following oligonucleotides were designed to target a sequence within the USP25 ORF: sh2962USP25fwd: AGCGCACAGAGGACATGATGAAGAATTAGTGAAGCCACAGATGTAATTCT CATCATGTCCTCTGTA, and sh2962USP25rev: GGCATACAGAGGACATGATGAAGAATTACATCT GTGGCTTCACTAATTCTTCATCATGTCCTCTGTG. The generated pSLIK-USP25-sh2962 (NEO) construct was sequence verified.

### 2.2. Cell Lines

HeLa cells (ATCC) were grown in GlutaMAX™-Minimum Essential Medium (MEM, Gibco, Thermo Fisher Scientific, Waltham, MA, USA), supplemented with 10% Fetal Bovine Serum High Performance (South American from Thermo Fisher Scientific, Waltham, MA, USA), 1 mM sodium pyruvate (Euroclone, Pero, Italy) and non-essential amino acids (Euroclone, Pero, Italy). For EGF-induced EGFR degradation, cells were serum starved overnight and then stimulated with human EGF. Stable inducible HeLa USP25 knock-down cells were produced by lentiviral infection using the pSLIK-USP25-sh2962 construct. After infection, selection of infected cells was performed by adding 400 μg/mL neomycin. USP25 knock-down was inducing by adding doxycycline to the medium (0.5 μg/mL final concentration).

### 2.3. siRNA Transfection

Transient knock-downs were performed using Stealth siRNA oligonucleotides from Thermo Fischer Scientific (Waltham, MA, USA). Cells were transfected twice by using RNAiMax (Invitrogen), first in suspension and the following day in adhesion. The following RNAis were used:USP25S1 CAGGAGGAGACAACUUACUACCAAA;AMSHS1 CGCUCUGGAGUUGAGAUUAUCCGAA;USP8S1 UCGUGAUGAGGAAAGGGCCUAUGUA.

### 2.4. Biochemistry

Cell lysis was performed in RIPA buffer (50 mM Tris-HCl, 150 mM NaCl, 1 mM EDTA, 1% Triton X-100, 1% sodium deoxycholate, and 0.1% SDS) supplemented with a protease inhibitor cocktail (Calbiochem) and the DUB inhibitor PR-619 (Calbiochem, MERCK, Darmstadt, Germany). The EGFR immunoblots were quantified using the ImageJ software, and results were expressed as percentage of EGFR over EGFR at time zero (T0). In all experiments, densitometry analyses were performed on different exposures of the blots and results were obtained in the linear phase of the exposure.

For immunoprecipitation experiments, 500 μg of total cell lysate in RIPA buffer were used. For co-immunoprecipitation experiments, 1 mg of total cell lysate in JS buffer (50 mM HEPES pH 7.5, 50 mM NaCl, 1% glycerol, 1% Triton X-100, 1.5 mM MgCl, and 25 mM EGTA) supplemented with a protease inhibitor cocktail (Calbiochem) were used. Lysates were incubated with the respective primary antibody for 1 h at 4 °C, prior to the addition of rec-protein G-Sepharose beads (Invitrogen, Thermo Fisher Scientific, Waltham, MA, USA) for 1 h at 4 °C. After extensive washes with lysis buffer (RIPA or JS), the beads were re-suspended in Laemmli buffer, boiled, and the proteins were analyzed through sodium dodecyl sulfate-polyacrylamide gel electrophoresis (SDS-PAGE) and immunoblotting.

### 2.5. ELISA-Based Dissociation-Enhanced Lanthanide Fluorescence Immunoassay (DELFIA)

We used the DELFIA technology from Perkin Elmer. Microwell plates were prepared with the capture antibody (rabbit polyclonal anti-EGFR 806, home-made, directed against aa 1172–1186 of human EGFR 5 μg/mL). Blocking was performed for 2 h with 2% BSA in PBS. A total of 25 μg of lysate, stimulated with 100 ng/mL of EGF, was incubated for 1 h at room temperature. After three washes, the wells were incubated with anti-EGFR (mouse monoclonal m108 diluted at 1 μg/mL in assay buffer), for 1 h at room temperature. After three washes, anti-mouse Europium-labelled secondary antibody (1 μg/mL in assay buffer) was added for an additional hour. After three washes and treatment with enhancement solution, the fluorescence was measured with EnVision instrument (Perkin Elmer, Waltham, MA, USA) (excitation at 340 nm and emission at 615 nm).

### 2.6. Immunofluorescence

Cells were plated on glass coverslips. For the internalization assay, cells were serum-starved for 4 h and incubated with Alexa555-EGF (40 ng/mL) and/or 13A9 antibody (20 μg/mL) for 1 h at 4 °C and shifted to 37 °C for various timepoints to allow internalization. The cells were then fixed in 4% paraformaldehyde (PFA) in PIPES buffer (0.2% BSA, 0.1% Triton X-100, and 1× PBS) for 10 min at room temperature. PFA-fixed cells were permeabilized with 0.1% Triton X-100 in 1× PBS for 5 min at room temperature. The cells were then incubated with 2% BSA in 1× PBS for 30 min at room temperature, followed by incubation for 30 min with Alexa488-conjugated donkey anti-mouse secondary antibody (A21202, Thermo Fisher Scientific, Waltham, MA, USA). The nuclei were DAPI-stained. The coverslips were analyzed using a confocal microscope (CLSM TCS SP8 STED, Leica), and the images were further processed with the ImageJ software.

### 2.7. EGF Internalization Assay and Measurement of Surface EGFRs

Internalization of ^125^I-EGF was performed at low EGF (1 ng/mL) and high EGF (20 ng/mL). All experiments were performed at 37 °C. HeLa cells were plated in 24-well plates (100.000/well) in triplicates for each time point and in an additional well for the determination of unspecific binding. The next day, cells were serum-starved for 4 h in binding buffer (serum-free medium supplemented with 0.1% BSA, 20 mM Hepes) and then incubated at 37 °C in the presence of 1 ng/mL ^125^I-EGF or 20 ng/mL EGF (1.5 ng/mL ^125^I-EGF + 18.5 ng/mL cold EGF) in 300 μL binding buffer (MEM, BSA 0.1%, Hepes pH 7.4 20 mM). At different time points (3, 5 and 7 min), cells were pit on ice, washed three times in ice-cold PBS, and then incubated for 5 min at 4 °C in 300 μL acid wash solution, pH 2.5 (acetic acid 0.2 M, NaCl 0.5 M). The radioactivity present in the acid wash represents the amount of ^125^I-EGF bound to the receptor on the cell surface. Cells were then lysed with 300 μL 1M NaOH—these samples represent the amount of internalized ^125^I-EGF. The unspecific binding was measured at each time point in the presence of an excess of non-radioactive EGF (300 times). After correction for non-specific binding, the ratio between internalized and surface-bound radioactivity was determined and plotted over time. Internalization rate constants (Ke) were extrapolated from the slopes of the trend line [28].

The number of surface-bound EGFRs was measured by ^125^I-EGF saturation binding as described in [11]. Briefly, serum-starved HeLa cells were incubated on ice for 6 h in the presence of EGF (100 ng/mL, spiked with ^125^I-EGF, typically in a 1:20 hot/cold ratio) in growth medium supplemented with 0.1% BSA and 20 mM HEPES pH 7.4. The cells were then washed three times with ice-cold PBS and solubilized in 1 M NaOH. After correction for the hot/cold dilution, the number of EGFRs/cell was deduced from the total recovered counts, corrected for the specific activity of the radioligand and divided with the number of cells in the plate. Non-specific binding was measured in the presence of a 200-fold excess of cold ligands and subtracted from the total counts.

### 2.8. TCGA Cohort’s Analysis

The following cancer patient databases belong to the Pan-Cancer Atlas (Hoadley et al., 2018, PMID: 29625048), which is derived from The Cancer Genome Atlas (TCGA) Research Network (https://www.cancer.gov/tcga): pancreatic adenocarcinoma, lung adenocarcinoma, lung squamous cell carcinoma, brain lower grade glioma (Brain LGG) colorectal adenocarcinoma and head and neck squamous cell carcinoma (HNSCC). For each database, patient samples with available RNAseq data were selected. USP25 and EGFR mRNA levels were analyzed using the cBioportal tool (https://www.cbioportal.org/) [29,30]. Normalized mRNA expressions derive from the z-scores relative to all samples (log RNA Seq V2 RSEM), performed with the cBioportal tool. In order to classify samples in EGFR- or USP25-high, mRNA expression higher than the median of the cohort was classified as high, otherwise the expression was classified as low. The graphical representation of the scatter dot plot was obtained through the GraphPad Prism (v.8). Complete data are reported below and shown the mean and standard deviation values of normalized USP25 mRNA expression for EGFR-high and -low groups. The *p*-values were derived from the Student’s *t*-test and the *q*-values were derived from the Benjamini–Hochberg procedure (Table 1).

## 3. Results

### 3.1. EGFR Degradation Occurs Faster upon USP25 Knock-Down

We previously identified USP25 as a potential regulator of EGFR turnover and fate [23]. To assess the role of USP25 in EGFR turnover, we first analyzed EGFR degradation upon USP25 siRNA transient knock-down (KD) in HeLa cells. Cells were stimulated with high-dose EGF (100 ng/mL) for different timepoints and the kinetics of EGFR degradation was assessed by immunoblot (Figure 1A) or dissociation-enhanced lanthanide fluorescence immunoassay (DELFIA) assay (Figure 1B). Enhanced EGFR degradation kinetics was observed upon USP25 KD. Indeed, depletion of USP25 caused a significant decrease in the level of EGFR, leaving only ~30% of the initial EGFR intact 2 h after the stimulation. Prompted by this result, we established a doxycycline inducible system to promote USP25 depletion in a stable model. Similar to the siRNA transient KD, we observed an accelerated EGFR degradation upon USP25 inducible depletion (Figure 1C).

USP25 activity has been involved in both proteasome-dependent and -independent signaling pathways [31,32]. In order to determine which degradative pathway is involved in the accelerated EGFR degradation observed upon USP25 depletion, we treated mock and USP25 KD cells with the lysosome inhibitor Chloroquine (CQN) or the proteasome inhibitor MG132 before EGF stimulation. CQN treatment (Figure 1D), but not MG132 treatment (Figure 1E), strongly inhibited EGFR degradation both in control and USP25 KD cells, indicating that depletion of USP25 causes an enhanced degradation of the EGFR via the lysosomal pathway.

### 3.2. EGFR Internalization Is Enhanced upon USP25 Knock-Down

Based on the accelerated EGFR degradation, we hypothesized that a faster trafficking of the EGFR occurs upon USP25 depletion. Thus, we first examined EGFR endocytosis at the single cell level, following both the fluorescently-labelled ligand and the EGFR, using an anti-EGFR antibody that recognizes the extracellular portion of the receptor without interfering with ligand binding or internalization [33]. As positive controls, we used AMSH and USP8, whose depletions affects EGFR trafficking with opposite effects [18,24]. At the early 3 min time point, the majority of the EGFR-EGF complexes resided at the plasma membrane in control cells, while in USP25 KD cells a substantial portion of the complexes was already internalized (Figure 2A). This phenotype appears to be similar to the one of ASMH KD [18]. As previously reported, USP8 KD cells showed an opposite behavior as a significant fraction of EGF/EGFR remained stuck in the endocytic compartments even at 2 h after stimulation [24]. Treatment with Chloroquine or MG132 do not affect the accelerated EGFR endocytosis visible in the USP25 KD cells (Figure 2B), indicating that the anticipated EGFR degradation possibly depends on a faster internalization kinetic.

To formally prove this idea, we decided to use a quantitative assay and monitor EGFR internalization with the radioactively labelled ligand ^125^I-EGF [10,28]. To determine if USP25 depletion impacts the CME and/or NCE pathways, we assessed the internalization rate at low-dose EGF (1 ng/mL) when only CME is active and at high-dose EGF (20 ng/mL) when NCE is the predominant EGFR endocytic pathway [6,10]. HeLa cells were serum-starved, followed by incubation with ^125^I-EGF for different timepoints and determination of EGFR internalization rate. At low-dose EGF, no significant difference was observed in EGFR internalization rates between control and USP25-depleted cells (Figure 3A), indicating that the CME pathway is not affected. A prerequisite to evaluate NCE in different cell lines is that the number of the EGFR at the cell surface should be stable [34]. Saturation binding assay enabled us to establish that the receptor number was not significantly affected by USP25 depletion (Figure 3B). Remarkably, at high-dose EGF, EGFR internalization rate was enhanced in USP25-depleted cells with a nearly twofold increase in the Ke compared with control cells (Figure 3C). Thus, USP25 depletion impacts on EGFR internalization when the NCE pathway is active. Importantly, we observed the same effect with the USP25 inducible KD system. To perform experiments in a homogeneous genetic background, we selected three clones that display a comparable expression of the receptor at the single cell level. Strikingly, an enhanced internalization rate at high dose of EGF was observed in all of them (Figure 3D).

These data indicate that USP25 is acting at the early steps of the NCE pathway, and that its depletion raises the EGFR internalization rate and, as a consequence, induces a faster degradation of the receptor. 

### 3.3. EGFR Ubiquitylation Is Accelerated upon USP25 Knock-Down

Since EGFR ubiquitylation is critical for NCE-mediated EGFR internalization [10,34], we sought to determine EGFR ubiquitylation status and dynamics upon USP25 transient KD. EGFR was immunoprecipitated from control and USP25-depleted cells stimulated with high-dose EGF (100 ng/mL) for different timepoints (Figure 4A). While in control cells EGFR ubiquitylation peaks at 10 min after EGF stimulation, in USP25-depleted cells the maximum level of EGFR ubiquitylation was observed already after 3 min of stimulation. The same kinetics were observed in the USP25 inducible KD system (Figure 4B). Prompted by this observation, we evaluated the ubiquitylation status of EGFR in the three selected clones at 3 min post induction. Our data confirmed that, in the absence of USP25, EGFR appears to be more ubiquitylated (Figure 4C).

Ubiquitylation of the EGFR displays a sharp threshold effect as a function of EGF concentration [11] and is sensitive to upward or downward modulation of c-Cbl, a rate-liming component of the process [12]. Therefore, we decided to evaluate the effect of USP25 depletion at different EGF concentrations at 3 min after stimulation. As observed in Figure 4D, an increased number of EGFRs modified with ubiquitin was detected in USP25-depleted cells compared with the control in all tested conditions, suggesting that the receptor is more responsive to c-Cbl-mediated ubiquitylation in the absence of USP25.

Upon EGF stimulation, a network of proteins becomes ubiquitylated [35], among which the endocytic machinery appears relevant for EGFR trafficking and degradation. In particular, ubiquitylation of the endocytic adaptor Eps15 is already induced at low doses of EGF stimulation and occurs early during the endocytic pathway [23,36]. We have previously reported that the ubiquitylation of Eps15 is counteracted by the activity of USP9X, with implications on EGFR internalization and trafficking [23]. Thus, we analyzed the status of Eps15 in USP25-depleted cells. We found that the ubiquitylation of Eps15 did not show any altered behavior compared to control cells (Figure 4E), suggesting that USP25 acts specifically at the level of the EGFR and is not a general regulator of ubiquitylated proteins involved in the early steps of EGFR internalization.

### 3.4. EGFR Phosphorylation Sites and c-Cbl Interaction Are Altered upon USP25 Knock-Down

At the molecular level, the efficient recruitment of c-Cbl to the active EGFR depends on the concomitant phosphorylation of a direct c-Cbl binding site, the Y1045, and two indirect sites through GRB2, the Y1068 and Y1086 in the EGFR tail [11]. Thus, we wondered whether the signaling events that lead to EGFR ubiquitylation are affected by USP25 depletion, and set out to assess the status of these specific phosphorylation sites in control and USP25 KD cells. Looking at the overall tyrosine phosphorylation status of the EGFR no difference was observed in USP25-depleted cells (Figure 5A), while both specific phosphorylation sites appeared affected. pY1068 was slightly increased whereas, surprisingly, pY1045 was consistently reduced with respect to control cells (Figure 5A,B). This result is counterintuitive as this EGFR phosphorylation event is required for c-Cbl-mediated EGFR ubiquitylation [11,13].

Next, we analyzed the interaction between the EGFR and c-Cbl in control and USP25-depleted cells by co-immunoprecipitation experiments. Interestingly, higher levels of c-Cbl co-immunoprecipitated with the EGFR in USP25-depleted cells compared to control cells (Figure 5C,D). This effect was also partially visible for the adaptor GRB2 (Figure 5C,D).

Two potential scenarios emerge from the data obtained: USP25 may regulate EGFR ubiquitylation by interacting with c-Cbl and limiting its binding to the receptor or, alternatively, by binding the receptor, USP25 may protect it from c-Cbl-mediated ubiquitylation. We tested both hypotheses by co-immunoprecipitation experiments in various conditions but did not observe a significant interaction of endogenous USP25 with c-Cbl or the EGFR. This latter result is not surprising, as the interaction between an enzyme and its target is labile by nature. Thus, we tried an overexpression approach using both GFP-USP25 wildtype (WT) or catalytically inactive mutant (C178A). Transfected HeLa cells were stimulated for 3 min with a high dose of EGF, and EGFR was immunoprecipitated with an anti-EGFR antibody or matched immunoglobulins, and USP25 co-immunoprecipitation was assessed with anti-USP25 antibody (Figure 5E). A slight signal corresponding to both GFP-USP25 WT and mutant was detected, suggesting that the interaction may be indirect. Interestingly, when we then looked at the effect of ectopically expressed GFP-USP25 WT on the EGFR internalization, we saw a reduction in the cytoplasmatic dots and a clear EGF signal decorating the plasma membrane even at 10 min after stimulation (Figure 5F). This impairment in EGFR endocytosis depends on the catalytic activity of USP25, as USP25 C178A had no detectable effect (Figure 5F).

Altogether, our results suggest that USP25, by affecting c-Cbl binding to the EGFR, could protect the receptor from c-Cbl-mediated ubiquitylation. In conditions of USP25 depletion, ubiquitylation is favored, with subsequent anticipated EGFR internalization and degradation.

### 3.5. USP25 and EGFR Expression Levels Correlate in Cancer Patients

Our data suggest that the levels of USP25 and EGFR expression may be connected. To establish the clinical relevance of our findings, we took advantage of publicly available cancer patient databases and analyzed the mRNA expression levels of USP25 and how they are linked to EGFR. In particular, we focused our attention on different types of cancers that are characterized by deregulated EGFR signaling [3]. Interestingly, we observed that USP25 mRNA levels are significantly higher in patients with stronger EGFR expression compared with patients with lower EGFR levels in all the tested tumor types (Figure 6A), reinforcing the concept of a functional connection between EGFR and USP25. Pancreatic adenocarcinoma shows the strongest upregulation of USP25 expression in tumors with a high EGFR expression, and this prompted us to evaluate cancer patient survival. Importantly, analysis of patient populations separated into prognosis-distinguishable subgroups showed that patients whose tumors exhibit high expression levels of both EGFR and USP25 have a lower overall survival compared with all the other patients (Figure 6B). It would be important to confirm this analysis looking at the protein expression level of the two proteins that do not necessarily behave as the corresponding mRNAs. Certainly, these results provide a promising rationale for future USP25-targeted strategies to improve therapeutic outcomes of poor prognosis subtypes.

## 4. Discussion

The EGFR-dependent signaling pathway represents a central molecular network in cellular physiology, controlling key processes such as proliferation and migration. To orchestrate suitable cellular responses, the EGFR is subjected to a tight regulation that is frequently altered in various human cancers. Ubiquitylation is a key signal in the EGFR pathway. By targeting the receptor and the endocytic machinery, ubiquitin mediates EGFR internalization and degradation via the NCE mechanism, leading to signal termination. Here, we characterize a DUB, USP25, which appears to control and fine-tune the internalization of the EGFR and its lysosomal degradation by modulating the amplitude of ubiquitylation on the receptor.

Our results suggest that USP25 directly modulates EGFR ubiquitylation by acting at the level of the receptor at early timepoints after EGF stimulation. Indeed, in the absence of USP25, the EGFR ubiquitylation dynamics is altered and the receptor reaches the highest level of ubiquitylation earlier than control after EGF stimulation. Moreover, we scored a slight but reproducible increase in the physical interaction between c-Cbl and the activated EGFR in the absence of USP25, suggesting that the depletion of USP25 could favor EGFR/c-Cbl interaction. While a physical competition is unlikely, USP25 could functionally compete with c-Cbl by deubiquitinating the receptor, hence limiting the amount of ubiquitylated EGFR and its consequent degradation. These results are in good agreement with previous findings showing that blocking c-Cbl binding to the EGFR is enough to inhibit EGFR internalization by the NCE pathway [11], and that both c-Cbl and EGFR concentrations are determinants for the cellular activation and extension of the pathway [12]. How the reduced phosphorylation of Y1045 observed upon USP25 depletion fits into this picture remains to be determined.

USP25 has been associated with different signaling pathways that are activated by distinct stimuli including IL-17-mediated inflammation response [31], virus- and pathogen-induced innate immune response [32,37], and Wnt signaling [38]. In all these cases, USP25 depletion affects the output of the stimulus, indicating that USP25 acts as a regulator of signaling pathways. We propose here that USP25 acts as a “security guard” of stimulus-associated ubiquitin-mediated signaling pathways to control the correct balance response after stimulation. Whether USP25 functions directly by deubiquitinating the signaling receptors in all these cases remains to be determined.

Previous studies provided evidence that elevated levels of USP25 are linked to the progression of different cancer types such as non-small-cell lung carcinoma (NSCLC) [39], which are characterized by EGFR genetic alterations and aberrant EGFR signaling. Consistent with this conclusion, we observed a positive correlation between USP25 and EGFR expression in a panel of cancer types. Moreover, pancreatic adenocarcinoma cancer patients with high levels of both USP25 and EGFR have the worse overall survival, suggesting that the USP25/EGFR axis is of potential interest for therapeutic intervention. Certainly, stabilization of activated EGFR by USP25 is likely to be an additional mechanism at the basis of the oncogenicity of the EGFR.

## 5. Conclusions

Understanding the molecular events that control the attenuation of the EGFR signal constitutes the key for its management in case of alteration. We have previously showed that binding between the receptor and c-Cbl is the major molecular event that initiates the NCE process [11,12]. We demonstrated that limited c-Cbl availability determines the uncoupling of EGFR phosphorylation and ubiquitylation at the supraphysiological EGFR level occurring in cancer [12]. Here, we add a new player in this game by showing that USP25 depletion increases EGFR ubiquitylation and degradation.

DUBs are emerging as promising drug targets and considerable efforts have been undertaken to develop specific DUB inhibitors [40]. Our findings provide an additional target to explore in the development of new therapeutic strategies for the treatment of tumors that depend on the EGFR signaling pathway, such as NSCLCs.

## Figures and Tables

**Figure 1 biomolecules-10-01548-f001:**
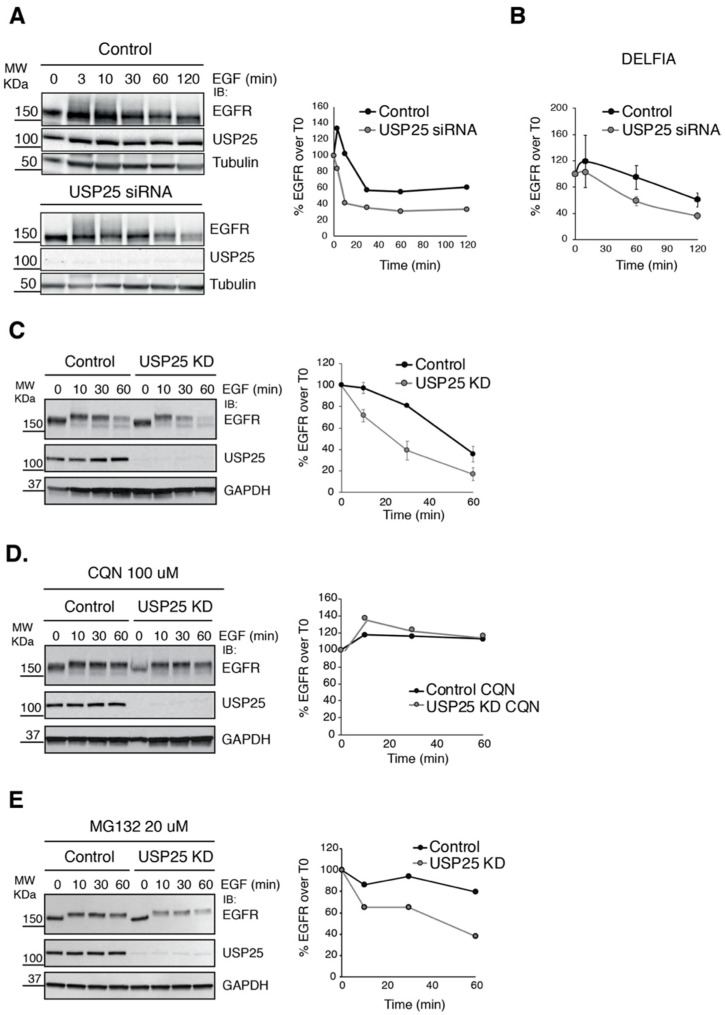
USP25 depletion induces a faster degradation of the EGFR. (**A**,**B**) Immunoblot (IB) and DELFIA analyses of HeLa cells transiently transfected with a RNAi oligo targeting USP25 or a scramble oligo (control). Serum-starved cells were stimulated with EGF (100 ng/mL), as indicated, and total cell lysates were analyzed by IB (**A**, left panel, one representative experiment of three is shown) and DELFIA (**B**) as indicated. (**C**, left panel) IB analysis, as in A, of a stable HeLa cell line carrying a shRNA targeting USP25 and control cells. USP25 depletion was induced with doxycycline treatment (0.5 μg/mL). One representative experiment of three is shown. (**D**,**E**, left panels) IB analysis of cells as in C, but pre-treated with Chloroquine (CQN 100 μM) for 1 h (**D**) or MG132 (20 μM) for 2 h (**E**) before stimulation with EGF. (**A**,**C**–**E**) Right panels: quantification of percentage EGFR signal, using the immunoblots.

**Figure 2 biomolecules-10-01548-f002:**
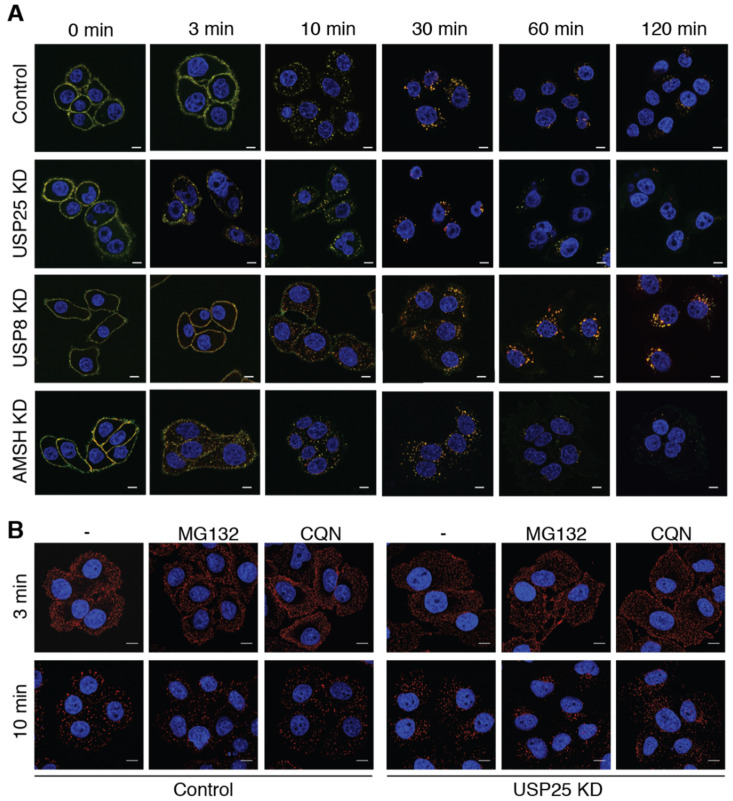
USP25 depletion accelerates EGFR internalization and trafficking. (**A**) IF analysis of HeLa cells transiently transfected with scramble (control), USP25, USP8, or AMSH RNAi oligos. Cells were serum-starved and stimulated with EGF as indicated in Materials and Methods. EGF, red; EGFR, green; DAPI, blue. Scale bar: 10 μm. EGFR internalization is accelerated in USP25 KD cells (see yellow puncta at the 3 min timepoint). (**B**) Cells treated as in A, were previously incubated with MG132 (20 mM for 1 h) or Chloroquine (CQN 100 mM for 1 h) before stimulation with Alexa555-EGF for the indicated timepoints. EGF, red; DAPI, blue. Scale bar 10 μm.

**Figure 3 biomolecules-10-01548-f003:**
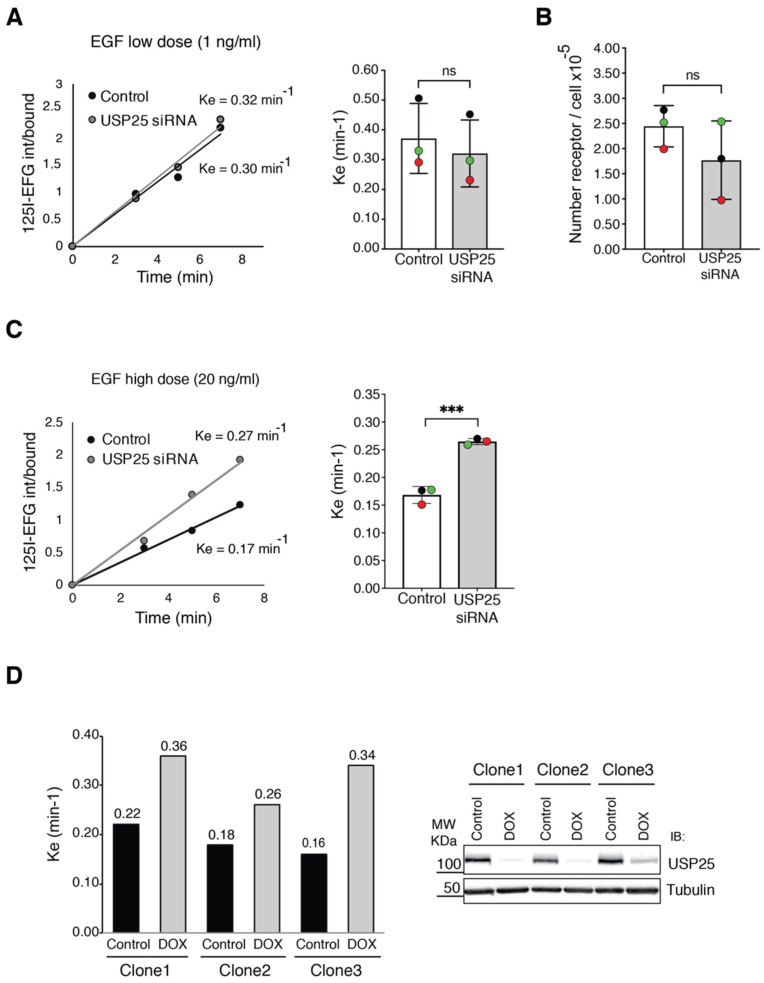
EGFR internalization is enhanced upon USP25 depletion. (**A**,**C**) Assessment of EGFR internalization in HeLa cells transiently transfected with a RNAi oligo targeting USP25 or a scramble oligo (control). Serum-starved cells were stimulated with low-dose (1 ng/mL) (**A**) or high-dose (20 ng/mL) radiolabeled EGF (^125^I-EGF) (**C**) as indicated. Results are expressed as the ratio between internalized and bound ligand. A representative internalization curve is shown (left panels). The Ke is the average of three independent experiments (each replicate is represented by a different color) ± SD. ns, not significant; *** *p* < 0.001 (right panel). (**B**) Saturation binding assay using ^125^I-EGF to determine the number of EGFR molecules on the cell surface. Results are an average of three independent experiments (each replicate is represented by a different color). (**D**) Left panel: assessment of EGFR internalization (at early timepoints, 0 and 7 min) in three different USP25 shRNA inducible clones. USP25 depletion was induced with doxycycline treatment (0.5 μg/mL). Right panel: validation of USP25 depletion by IB analysis of total cell lysates as indicated.

**Figure 4 biomolecules-10-01548-f004:**
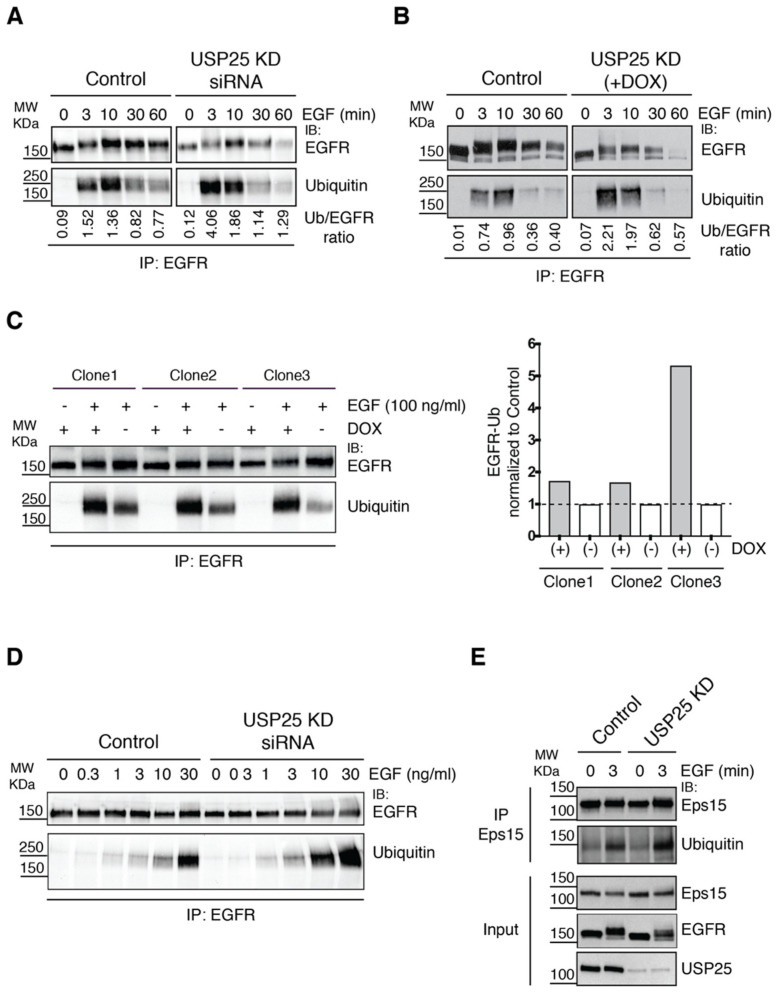
EGFR ubiquitylation dynamics is affected by USP25 depletion. (**A**) Immunoprecipitation (IP) and IB analyses of HeLa cells transfected with a RNAi oligo targeting USP25 or a scramble oligo (control). Serum-starved cells were stimulated with EGF (100 ng/mL) as indicated. 500 μg of cell lysates were IP and analyzed by IB as indicated. (**B**) IP and IB analyses, as in A, of a stable HeLa cell line carrying a shRNA targeting USP25 and control cells. USP25 depletion was induced with doxycycline treatment (0.5 μg/mL). (**C**) IP and IB analyses, as in B, of EGFR ubiquitylation after 3 min of EGF stimulation (100 ng/mL) in three different HeLa-USP25 shRNA inducible clones. USP25 depletion was induced with doxycycline treatment (0.5 μg/mL). Right panel: quantification of Ubiquitylated EGFR normalized to control. The dotted line indicates the value of the control. (**D**) IP and IB analyses, as in B, using different EGF concentrations, as indicated. (**E**) IP and IB analyses, of cells as in B, using an anti-Eps15 antibody. Input (50 μg) corresponds to 10% of total immunoprecipitated proteins.

**Figure 5 biomolecules-10-01548-f005:**
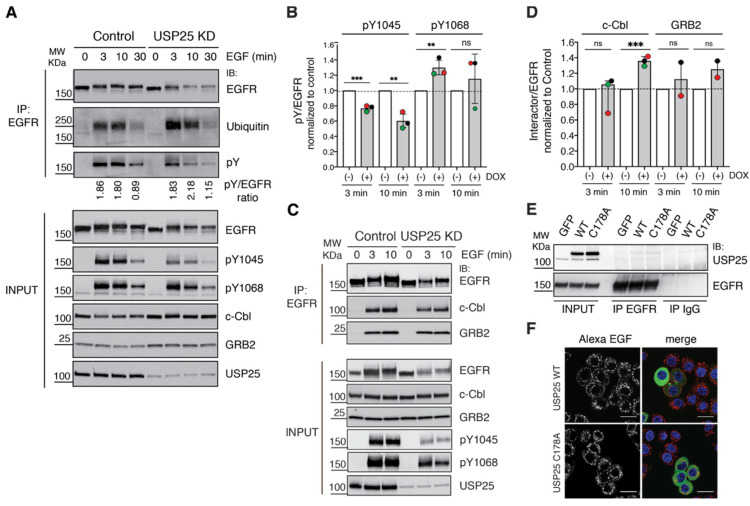
Altered phosphorylation and c-Cbl-EGFR interaction upon USP25 depletion. (**A**) Biochemical analysis of HeLa cells stably transfected with a shRNA targeting USP25 (USP25 KD) and control cells. USP25 depletion was induced with doxycycline treatment (0.5 μg/mL). Serum-starved cells were stimulated with EGF (100 ng/mL) and total cell lysates were prepared. A total of 500 μg of cell lysates were IP and analyzed by IB as indicated. (**B**) Quantification of pY1045 and pY1068 sites, pY1045/EGFR and pY1068/EGFR ratios were calculated and normalized to control. Results are an average of three independent experiments ± SD, ns, not significant, ** *p* < 0.01, *** *p* < 0.001 (replicates are color coded). (**C**) Biochemical analysis of HeLa cells treated as in A. A total of 1 mg of cell lysates were IP and analyzed by IB as indicated. Input corresponds to 10% of total immunoprecipitated proteins. (**D**) Quantification of c-Cbl and GRB2 interaction with EGFR. C-Cbl/EGFR and GRB2/EGFR immunoprecipitated ratios were calculated and normalized to control. Results are an average of three (c-Cbl) or two (GRB2) independent experiments ± SD, *** *p* < 0.001 (replicates are color-coded). (**E**) HeLa cells transfected with GFP-USP25 WT or GFP-USP25 C178A were serum starved and stimulated for 3 min with high-dose EGF (100 ng/mL). A total of 1 mg of cell lysates were IP and analyzed by IB as indicated. Input corresponds to 5% of total immunoprecipitated proteins. (**F**) HeLa cells transfected as in E, were serum-starved and then stimulated with Alexa555-EGF for the indicated timepoints. EGF, red; DAPI, blue. Scale bar 10 μm.

**Figure 6 biomolecules-10-01548-f006:**
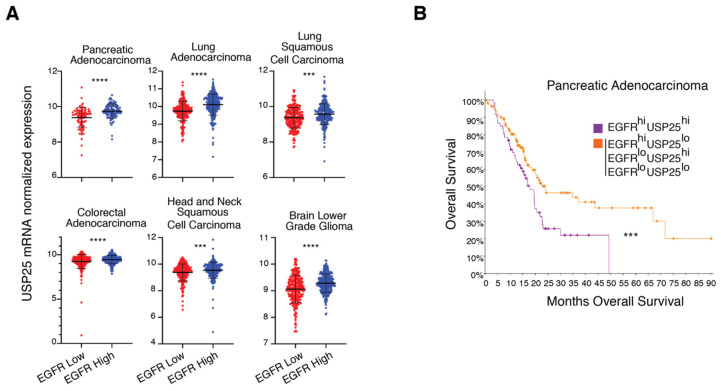
High levels of USP25 correlate with EGFR expression and induce a worse overall survival in cancer patients. (**A**) Scatter dot plot of USP25 mRNA expression in cancer patients with high (blue) or low (red) EGFR mRNA expression. Patient samples were classified as EGFR high if their normalized EGFR mRNA expression value was higher than the median value. The following cancer types were analyzed: colorectal adenocarcinoma (N = 592), lung adenocarcinoma (N = 510), lung squamous cell carcinoma (N = 488), pancreatic adenocarcinoma (N = 177), head and neck squamous cell carcinoma (N = 515), and brain lower grade glioma (N = 514). **** *q*-value < 0.0001, *** *q*-value < 0.001 by Benjamini–Hochberg procedure. (**B**) Kaplan–Meier curve of overall survival for pancreatic adenocarcinoma patients according to their EGFR and USP25 mRNA expression. Patient samples were classified as EGFR high (or USP25 high) if their normalized mRNA levels were higher than the median value. Patients with both EGFR high and USP25 high expression were clustered together (purple) and overall survival was compared with that of all the other patients (orange). *** *p*-value < 0.001 by logrank test. The data in A and B were analyzed through the cBioportal tool (https://www.cbioportal.org).

**Table 1 biomolecules-10-01548-t001:** Data obtained from the TCGA Cohort’s Analysis.

	USP25 Mean log2 in EGFR Low Group	USP25 Mean log2 in EGFR High Group	Standard Deviation in EGFR Low Group	Standard Deviation in EGFR High Group	*p*-Value	*q*-Value
Pancreatic Adenocarcinoma	9.38	9.74	0.55	0.40	1.248 × 10^−6^	1.875 × 10^−5^
Lung squamous cell carcinoma	9.37	9.57	0.58	0.59	2.678 × 10^−4^	1.622 × 10^−3^
Lung Adenocarcinoma	9.74	10.11	0.55	0.60	5.61 × 10^−13^	3.00 × 10^−11^
Brain lower grade glioma	9.06	9.29	0.52	0.35	1.91 × 10^−8^	1.44 × 10^−7^
Colorectal Adenocarcinoma	9.24	9.46	0.79	0.47	5.271 × 10^−5^	2.231 × 10^−4^
Head and neck squamous cell carcinoma	9.37	9.54	0.64	0.60	2.440 × 10^−3^	5.722 × 10^−3^

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
