# Peer review of "USP25 Regulates EGFR Fate by Modulating EGF-Induced Ubiquitylation Dynamics"

_biomolecules, 2020, doi:10.3390/biom10111548_

Round 1

Reviewer 1 Report

This manuscript written by Nino et al. reports on their discovery that USP25 modulates EGFR signaling.  Building on their previous studies on EGFR regulation, the authors provide a clear rationale for their experimental approach to characterize the role of USP25 in competing with c-Cbl ubiquitylation of EGFR.  Using RNAi knockdowns, immunoprecipitation, DELFIA, immunofluorescence, and EGF internalization assays, the authors experimentally support their assertion that USP25 plays an important role in EGFR regulation.  The authors also analyzed cancer patient databases curated in the TCGA demonstrating that their observation that USP25/c-Cbl control of EGFR could serve as a good therapeutic target with clinical relevance.  The authors should be commended on writing such a clear and easy to follow manuscript that will undoubtedly be helpful to other cancer researchers currently studying the role of ubiquitylation in oncogenesis.

Suggestion: Since the attachment of ubiquitin to a protein is a posttranslational modification, the correct term that should be used throughout this manuscript is “ubiquitylation”.  This is analogous to phosphorylation, methylation, acetylation, acylation, etc.

Minor Typos:

  • Line 29 = inducing receptor
  • Line 103 = Thermo Fisher
  • Line 125 = 25 μg of lysate
  • Line 134 = For the internalization assay,
  • Line 155 = 300 μL
  • Line 176 = were analyzed using the cBioportal tool
  • Line 413 = kinetics of pY1068
  • Line 423 = We tested both hypotheses
  • Line 474 = 500 μg of cell

Author Response

Reviewer #1:

This manuscript written by Nino et al. reports on their discovery that USP25 modulates EGFR signaling.  Building on their previous studies on EGFR regulation, the authors provide a clear rationale for their experimental approach to characterize the role of USP25 in competing with c-Cbl ubiquitylation of EGFR.  Using RNAi knockdowns, immunoprecipitation, DELFIA, immunofluorescence, and EGF internalization assays, the authors experimentally support their assertion that USP25 plays an important role in EGFR regulation.  The authors also analyzed cancer patient databases curated in the TCGA demonstrating that their observation that USP25/c-Cbl control of EGFR could serve as a good therapeutic target with clinical relevance.  The authors should be commended on writing such a clear and easy to follow manuscript that will undoubtedly be helpful to other cancer researchers currently studying the role of ubiquitylation in oncogenesis.

  1. We sincerely thank the reviewer for his/her nice words.

Suggestion: Since the attachment of ubiquitin to a protein is a posttranslational  modification, the correct term that should be used throughout this manuscript is “ubiquitylation”.  This is analogous to phosphorylation, methylation, acetylation, acylation, etc.

  1. We thank the reviewer for bringing this to our attention. According to his/her suggestion we changed the word ubiquitination in ubiquitylation throughout the manuscript.

Minor Typos:

  • Line 29 = inducing receptor
  • Line 103 = Thermo Fisher
  • Line 125 = 25 μg of lysate
  • Line 134 = For the internalization assay,
  • Line 155 = 300 μL
  • Line 176 = were analyzed using the cBioportal tool
  • Line 413 = kinetics of pY1068
  • Line 423 = We tested both hypotheses
  • Line 474 = 500 μg of cell
  1. We apologize for these typos and grammatical errors. We have corrected all the sentences pointed out by the reviewer.

Reviewer 2 Report

I read the manuscript entitled "USP25 regulates EGFR fate by modulating EGF-2 induced ubiquitination dynamics" with great interest. This manuscript used a set of cell-based assays to show depletion of USP25 can alter EGFR ubiquitination resulting receptor internalization and degradation. Based on the provided results, authors suggested that USP25 may function as a “security guard” during EGFR internalization and degradation by the lysosome pathway. Finally, their analyzed human data suggests that USP25 can be a key element and consequently a potential therapeutic factor in cancer cells with elevated level of EGFR. The majority of results are satisfactory. Although, all biological assays have been conducted in a single cell line, HeLa cells. The manuscript has been well written. Figure legends are self-explanatory.  

Please find below a list of comments and concerns that I hope the authors will find helpful in improving this manuscript.

  • Authors used a few approaches in figure 4 to show that the absence of USP25 can lead to elevation of ubiquitinated EGFR. However, panels in Fig.4 just show a single band at 250kDa as a ubiquitinated EGFR. Is this a poly-mono-ubiquitinated EGFR? Did authors observed a ladder of ubiquitinated EGFR as previously described (PMID: 17940017). It would be more informative if authors can show all bands between 150kDa to 250kDa and above in corresponding panels in Fig 4.
  • It would be more convincing if author could include a set of HeLa cells transfected with GFP as a control group in Fig. 5C. Also, a mouse or rabbit immunoglobulin (serum) is more standard for an IP experiment than using a “non-related antibody”.
  • The TCGA cohort’s analysis presented in Fig. 6 is based on the RNA level of USP20 and EGFR. The level of RNA cannot directly explain the post-translational modification effect of USP25 on EGFR protein in tumor tissues. In fact, according to the Human protein Atlas, USP25 is not prognostic in pancreatic cancer. Authors can highlight that the protein levels of USP25 and EGFR can be more reliable to predict the overall survival of patients.
  • Does MG132 and/or CQN reverse accelerated EGFR internalization and trafficking in USP25 depletion illustrated in Fig. 2.
  • Why authors preferred MG132 to a more selective proteasome inhibitor such as bortezomib?

Author Response

Reviewer #2:

I read the manuscript entitled "USP25 regulates EGFR fate by modulating EGF-2 induced ubiquitination dynamics" with great interest. This manuscript used a set of cell-based assays to show depletion of USP25 can alter EGFR ubiquitination resulting receptor internalization and degradation. Based on the provided results, authors suggested that USP25 may function as a “security guard” during EGFR internalization and degradation by the lysosome pathway. Finally, their analyzed human data suggests that USP25 can be a key element and consequently a potential therapeutic factor in cancer cells with elevated level of EGFR. The majority of results are satisfactory. Although, all biological assays have been conducted in a single cell line, HeLa cells. The manuscript has been well written. Figure legends are self-explanatory.  

  1. We are very grateful to reviewer 2 for the critical reading of our manuscript and for his/her nice words.

Please find below a list of comments and concerns that I hope the authors will find helpful in improving this manuscript.

Authors used a few approaches in figure 4 to show that the absence of USP25 can lead to elevation of ubiquitinated EGFR. However, panels in Fig.4 just show a single band at 250kDa as a ubiquitinated EGFR. Is this a poly-mono-ubiquitinated EGFR? Did authors observed a ladder of ubiquitinated EGFR as previously described (PMID: 17940017). It would be more informative if authors can show all bands between 150kDa to 250kDa and above in corresponding panels in Fig 4.

  1. We thank the reviewer for having underline this point. As the reviewer mentioned, when analysed by immunoblot, the EGFR ubiquitination signal usually looks more like a smear/ladder that as a single band. This can be clearly observed in the paper mentioned by the reviewer (Huang et al PNAS, 2007, PMID: 17940017), but also in our previous publication (Fig. 5; Sigismund et al PNAS 2005, PMID 15701692). It is important to note that the EGFR-Ub signal pattern depends on the resolution of the acrylamide gel. In particular, low percentage of gels (7,5%) allow a better resolution of the high molecular weight markers. Vice versa, in the pre-cast gradient gels (4-12%), like the ones used in our current paper, the EGFR-Ub signal is more compact. We used these gels to better appreciate the differences between controls and USP25 depleted cells. Following the reviewer’s suggestion, we prepared larger images of the EGFR-Ub blots in order to show the region between 150KDa to 250KDa and above.

It would be more convincing if author could include a set of HeLa cells transfected with GFP as a control group in Fig. 5C. Also, a mouse or rabbit immunoglobulin (serum) is more standard for an IP experiment than using a “non-related antibody”.

  1. We thank the reviewer for this suggestion. As this point was raised also by reviewer n.3, we repeated the experiment including additional controls (new Fig. 5E). HeLa cells were transfected with either GFP alone, GFP-USP25-WT, or GFP-USP25-C178A and co-immunoprecipitation was performed in parallel with rabbit polyclonal anti-EGFR antibody or rabbit immunoglobulin as a control. In this new experiment the level of co-immunoprecipitation was much lower, raising issue on a direct physical interaction between the EGFR and USP25. Thus, we decided to tone down our conclusions and we have modified the text accordingly.

The TCGA cohort’s analysis presented in Fig. 6 is based on the RNA level of USP20 and EGFR. The level of RNA cannot directly explain the post-translational modification effect of USP25 on EGFR protein in tumor tissues. In fact, according to the Human protein Atlas, USP25 is not prognostic in pancreatic cancer. Authors can highlight that the protein levels of USP25 and EGFR can be more reliable to predict the overall survival of patients.

  1. We totally agree with the reviewer and we have now added a sentence to clarify the limit of our analysis (Results section 3,5, line 545).

Does MG132 and/or CQN reverse accelerated EGFR internalization and trafficking in USP25 depletion illustrated in Fig. 2.

  1. We thank the reviewer for this interesting observation. To answer his/her question, we performed EGF stimulation under CQN or MG132 treatments and analysed EGFR/EGF internalization by immunofluorescence by using confocal microscopy (new Fig. 2B). Neither CQN nor MG132 reversed the accelerated EGFR internalization observed in USP25 depleted cells. Thanks to this important addition, we can now conclude that the EGFR degradation rescue observed with CQN treatment of USP25 KD cells (Fig. 1D) is not due to the inhibition of the anticipate kinetic of EGFR internalization.

Why authors preferred MG132 to a more selective proteasome inhibitor such as bortezomib?

  1. The reason is merely economic. We agree with the reviewer that MG132 can inhibit not only the proteasome but calpains as well. However, the involvement of calpains is unlikely to be relevant in our system. Certainly, the effect observed with USP25 depletion could not be ascribed to the proteasome. To confirm the efficicacy of the proteasome inhibition in the experiment presented in Fig 5E, we analysed the stabilization of ß-catenin, a known proteasome target. Results are reported below and can be added to the figure if requested.

Reviewer 3 Report

The manuscript by Nino and co-workers describes a role for the deubiquitylation (DUB) enzyme UPS25 in EGFR internalization. The issue of EGFR turnover has been studied thoroughly for three decades and many cellular factors were demonstrated to partake in this mechanism. Some factors engage internalization, some are involved in trafficking, and some in lysosomal degradation. The authors found that UPS25 acts at the early steps of EGFR internalization by interfering with c-CBL binding, thereby reducing EGFR ubiquitylation. The study establishes UPS25 as a negative effector of EGFR downregulation. Overexpression of both EGFR and UPS25 correlates with poor prognosis in pancreatic adenocarcinoma patients, implicating it as a potential pharmacological target.

The manuscript is very clear and well written, and most of the data is substantial. Yet, it also raises several questions that need to be clarified prior to consideration for publication.

Major comments

  • The data in figure 5 is the key to understanding the role of UPS25 in EGFR internalization. However, the conclusion in lines 432-434 are not fully supported by the authors' findings for several reasons:
  • The authors observed only a “minor decrease in pY1045”, however according to my judgment, the decrease is at least 2-3 fold. This was done in cells in which UPS25 was only partially knocked down. The effect might be stronger in cells that do not express the DUB at all.
  • The authors observed “higher levels of c-Cbl co-immunoprecipitated with the EGFR in USP25-depleted cells”. However, according to my judgment, the effect of UPS25 on binding is smaller than its effect on Y1045 phosphorylation. This was done in cells in which UPS25 was strongly knocked down.
  • In Figure 5C, the authors overexpress active-site mutant UPS25 and show that it still binds EGFR. However, this experiment lacks several controls. (a) Binding of the mutant should be compared to the wild type, overexpressed to the same level. (b) The authors' interpretation implies that UPS25 activity is irrelevant to its effect on EGFR ubiquitylation and degradation. The authors should show it by demonstrating the same effects as the wild type protein.

Altogether, I am not convinced about the authors' conclusion that “USP25, by binding to the EGFR, could protect the receptor from c-Cbl-mediated ubiquitination”. I recommend performing both the phosphorylation and IP experiments in 5A and 5B on the same cell population and quantify them properly.

Minor comments

  • In their introduction, the authors ignore early key literature, mainly about the mechanism of EGFR internalization and the role of ubiquitylation (For example, Levkowitz et al., 1998, 19999 describing the role of Tyr1045 of the EGFR in recruiting c-Cbl).  
  • Figure 1, The authors demonstrate changes in EGFR levels, following EGF treatment, and refer to it as a degradation mechanism. However, as shown in Fig. 1A, EGFR levels can also increase, suggesting that changes in the steady-state levels involve changes in both the synthesis and degradation rates. Therefore, I suggest that the authors will use different terminology. For example, turnover instead of degradation. Alternatively, the authors can study degradation kinetics directly, for example by blocking overall protein synthesis using ribosome inhibitors or labeling surface receptors to follow their elimination rate.
  • Figure 4, Data interpretation could be more effective if the authors quantitate and normalize ubiquitylation/EGFR. Alternatively/additionally, the authors could keep EGFR levels constant, for example by shifting cells to low temperatures or using certain mutants to block internalization.

Author Response

Reviewer #3:

The manuscript by Nino and co-workers describes a role for the deubiquitylation (DUB) enzyme UPS25 in EGFR internalization. The issue of EGFR turnover has been studied thoroughly for three decades and many cellular factors were demonstrated to partake in this mechanism. Some factors engage internalization, some are involved in trafficking, and some in lysosomal degradation. The authors found that UPS25 acts at the early steps of EGFR internalization by interfering with c-CBL binding, thereby reducing EGFR ubiquitylation. The study establishes UPS25 as a negative effector of EGFR downregulation. Overexpression of both EGFR and UPS25 correlates with poor prognosis in pancreatic adenocarcinoma patients, implicating it as a potential pharmacological target.

The manuscript is very clear and well written, and most of the data is substantial. Yet, it also raises several questions that need to be clarified prior to consideration for publication.

  1. We are very grateful to Referee 3 for the critical reading of our manuscript and the numerous useful suggestions that we have now implemented to improve our study.

The data in figure 5 is the key to understanding the role of UPS25 in EGFR internalization. However, the conclusion in lines 432-434 are not fully supported by the authors' findings for several reasons:

The authors observed only a “minor decrease in pY1045”, however according to my judgment, the decrease is at least 2-3 fold. This was done in cells in which UPS25 was only partially knocked down. The effect might be stronger in cells that do not express the DUB at all.

  1. We apologize for not having conveyed the correct message and we agree with the reviewer that this is an important point. We have now quantified the results from three independent experiments, and normalized the pY1045 (and pY1068) signal respect the corresponding EGFR level that is lower in depleted cells compared to control. The pY1045/EGFR ratio quantified in USP25 depleted cells was then reported to the respective control. This quantitative analysis (included in the new Fig. 5B) shows that the decrease in pY1045 observed in USP25 depleted cells is reproducible and significant. For completeness, we also included the data of the total EGFR-pY, which also seems lower in USP25 depleted cells but is not.

The authors observed “higher levels of c-Cbl co-immunoprecipitated with the EGFR in USP25-depleted cells”. However, according to my judgment, the effect of UPS25 on binding is smaller than its effect on Y1045 phosphorylation. This was done in cells in which UPS25 was strongly knocked down

  1. We did not intend to exclude that the reduced Y1045 phosphorylation might contribute to the phenotype, but its immediate interpretation is counterintuitive. How the reduced phosphorylation of the Cbl binding site may increase EGFR ubiquitination remains an open question and we now stated this clearly in the discussion. We agree with the reviewer that the increased Cbl binding observed in USP25 depleted cells is not a major change (now quantified in the new Fig. 5D), but, similar to the other data we obtained, it is consistent with the idea of a counteracting activity between Cbl and USP25. Following the reviewer’s criticism, we toned down our conclusions to avoid misinterpretations. We are ready to do it even further if the reviewer thinks that this is advisable.

In Figure 5C, the authors overexpress active-site mutant UPS25 and show that it still binds EGFR. However, this experiment lacks several controls. (a) Binding of the mutant should be compared to the wild type, overexpressed to the same level.

  1. As mentioned before, we repeated the experiment including additional controls (see detailed reply to point 3 of Rev. 2).

(b) The authors' interpretation implies that UPS25 activity is irrelevant to its effect on EGFR ubiquitylation and degradation. The authors should show it by demonstrating the same effects as the wild type protein.

  1. We apologize for not being clear enough, allowing misleading interpretations. The use of the USP25 catalytic inactive mutant (C178A) was used as a strategy to optimize the interaction between the EGFR (substrate) and USP25. This system is commonly used to stabilize interaction between an enzyme and its substrate. Unfortunately, the obtained results show that the physical interaction between USP25 and EGFR is limited and not different between WT and C178A mutant. However, this result does not imply that the USP25 catalytic activity is irrelevant in the context of the EGFR. At the biochemical level, the limited % of transfection precludes the analysis of the ectopically express USP25 on the EGFR internalization and degradation. Taking this in consideration, we decided to analyse the effect of GFP-USP25 overexpression on the EGFR internalization at the single cell level by confocal microscopy. While GFP-USP25 WT caused a delay in EGFR internalization; overexpression of a catalytically inactive mutant instead showed no detectable effect on EGFR endocytosis, indicating that USP25 activity is essential for the assessed internalization defects. These data are now included in new Fig 5F.

Altogether, I am not convinced about the authors' conclusion that “USP25, by binding to the EGFR, could protect the receptor from c-Cbl-mediated ubiquitination”. I recommend performing both the phosphorylation and IP experiments in 5A and 5B on the same cell population and quantify them properly.

  1. To address the concern expressed by the Reviewer, we have now performed additional experiments testing both phosphorylation and co-IP at the same time. We also performed accurate quantification of three experiments that are reported together with the original pictures. We hope that our new results (Fig. 5 B,C,D) together with the changes added in the manuscript clarified the point raised.

Minor comments

In their introduction, the authors ignore early key literature, mainly about the mechanism of EGFR internalization and the role of ubiquitylation (For example, Levkowitz et al., 1998, 19999 describing the role of Tyr1045 of the EGFR in recruiting c-Cbl).  

  1. We apologize for this and we added additional relevant references (Levkowitz et al., 1999 PMID: 10635327; Waterman et al., 2002, PMID: 11823423; Jiang et al., 2003, PMID: 12631709).

Figure 1, The authors demonstrate changes in EGFR levels, following EGF treatment, and refer to it as a degradation mechanism. However, as shown in Fig. 1A, EGFR levels can also increase, suggesting that changes in the steady-state levels involve changes in both the synthesis and degradation rates. Therefore, I suggest that the authors will use different terminology. For example, turnover instead of degradation. Alternatively, the authors can study degradation kinetics directly, for example by blocking overall protein synthesis using ribosome inhibitors or labeling surface receptors to follow their elimination rate.

  1. We apologize for not having explained this important point in the original version of the manuscript. We agree with the reviewer that changes in the steady-state levels involve changes in both the synthesis and degradation rates. However, we measured surface levels of EGFR by 125I-EGF to find no significant changes between control and USP25 depleted cells. Moreover, results we obtained by immunoblot and DELFIA were always expressed as ‘% of initial EGFR’, which means that they were normalized to the numerical value at time 0 (that may vary between the different cells). Finally, measuring the inhibition of lysosomal activity obtained with the Chloroquine treatment fully rescued the USP25 phenotype indicating that the decrease in EGFR levels observed in our experiments is definitively due to degradation.

Figure 4, Data interpretation could be more effective if the authors quantitate and normalize ubiquitylation/EGFR. Alternatively/additionally, the authors could keep EGFR levels constant, for example by shifting cells to low temperatures or using certain mutants to block internalization.

  1. We thank the reviewer for his/her suggestion. We quantified the Ub/EGFR ratio and indicated the values below the respective blots.

Round 2

Reviewer 2 Report

Please fix below new inserted sentence (line 648) in section 3.5. as follow: "It would be important to confirm this analysis looking at the mRNA expression levels of the two genes that not necessarily behave as the corresponding proteins. Certainly, these results provide a promising …".

Reviewer 3 Report

The authors thoroughly addressed my concerns and provide solutions to most of the issues.